# Threonine Deficiency Increases Triglyceride Deposition in Primary Duck Hepatocytes by Reducing STAT3 Phosphorylation

**DOI:** 10.3390/ijms25158142

**Published:** 2024-07-26

**Authors:** Zhong Zhuang, Wenqian Jia, Lei Wu, Yongpeng Li, Yijia Lu, Minghong Xu, Hao Bai, Yulin Bi, Zhixiu Wang, Shihao Chen, Yong Jiang, Guobin Chang

**Affiliations:** 1Key Laboratory for Animal Genetics & Molecular Breeding of Jiangsu Province, College of Animal Science and Technology, Yangzhou University, Yangzhou 225009, China; zz150211@163.com (Z.Z.); 19515798050@163.com (W.J.); 17320275567@163.com (L.W.); 17794281505@163.com (Y.L.); yijialu2023@163.com (Y.L.); m18762323813@163.com (M.X.); ylbi@yzu.edu.cn (Y.B.); wangzx@yzu.edu.cn (Z.W.); mrrchen@yzu.edu.cn (S.C.); gbchang1975@yzu.edu.cn (G.C.); 2Joint International Research Laboratory of Agriculture and Agri-Product Safety, The Ministry of Education of China, Institutes of Agricultural Science and Technology Development, Yangzhou University, Yangzhou 225009, China; bhowen1027@yzu.edu.cn

**Keywords:** duck, threonine, STAT3, hepatocyte, triglyceride

## Abstract

Liver lipid metabolism disruption significantly contributes to excessive fat buildup in waterfowl. Research suggests that the supplementation of Threonine (Thr) in the diet can improve liver lipid metabolism disorder, while Thr deficiency can lead to such metabolic disorders in the liver. The mechanisms through which Thr regulates lipid metabolism remain unclear. STAT3 (signal transducer and activator of transcription 3), a crucial transcription factor in the JAK-STAT (Janus kinase–signal transducer and activator of transcription) pathway, participates in various biological processes, including lipid and energy metabolism. This research investigates the potential involvement of STAT3 in the increased lipid storage seen in primary duck hepatocytes as a result of a lack of Thr. Using small interfering RNA and Stattic, a specific STAT3 phosphorylation inhibitor, we explored the impact of STAT3 expression patterns on Thr-regulated lipid synthesis metabolism in hepatocytes. Through transcriptome sequencing, we uncovered pathways related to lipid synthesis and metabolism jointly regulated by Thr and STAT3. The results showed that Thr deficiency increases lipid deposition in primary duck hepatocytes (*p* < 0.01). The decrease in protein and phosphorylation levels of STAT3 directly caused this deposition (*p* < 0.01). Transcriptomic analysis revealed that Thr deficiency and STAT3 knockdown jointly altered the mRNA expression levels of pathways related to long-chain fatty acid synthesis and energy metabolism (*p* < 0.05). Thr deficiency, through mediating STAT3 inactivation, upregulated *ELOVL7*, *PPARG*, *MMP1*, *MMP13*, and *TIMP4* mRNA levels, and downregulated *PTGS2* mRNA levels (*p* < 0.01). In summary, these results suggest that Thr deficiency promotes lipid synthesis, reduces lipid breakdown, and leads to lipid metabolism disorders and triglyceride deposition by downregulating STAT3 activity in primary duck hepatocytes.

## 1. Introduction

To reduce feed costs, low-protein diets have been applied in the duck industry [1,2]. However, the application of low-protein diets often leads to lipid metabolism disorders [3,4], which affects the health of ducks, leads to excessive abdominal fat deposition [5], and reduces slaughter rates and carcass quality [6]. Previous studies showed that balancing amino acid levels in low-protein diets by amino acid supplementation in diets increases growth performance and reduces hepatic lipid deposition [7,8].

Threonine (Thr) is the third limiting amino acid (which cannot be synthesized in the body) in waterfowl, and plays a critical role in preventing liver lipid metabolism disorders and improving growth [9,10]. Our previous research has shown that increased dietary threonine levels can lower total lipid triglyceride (TG), cholesterol, and low-density lipoprotein cholesterol in the livers of meat ducks [11]. By contrast, dietary Thr deficiency has been found to increase hepatic TG deposition in Pekin ducks, and to impair liver function and growth performance [8]. Lipid metabolism disorder is characterized by an excessive accumulation of lipids in the liver [12]. It results from an imbalance between liver lipid synthesis, transport, and degradation [13,14]. However, the specific metabolic pathways and molecular mechanisms by which Thr regulates lipid deposition in waterfowl livers remain unknown.

The Janus kinase–signal transducer and activator of the transcription signaling pathway (JAK-STAT) is vital for regulating lipid metabolism [15,16,17]. STAT3, an essential transcription factor in the JAK-STAT signaling pathway, is highly expressed in the liver and involved in lipid metabolism. Upon activation by cytokines, phosphorylated STAT protein enters the nucleus to regulate gene transcription [18,19]. Treating obese mice with STAT3-inducing cytokines (interleukins IL-6 and IL-22), or enhancing STAT3 expression, reduces hepatic lipid deposition caused by a high-fat diet [20,21]. Conversely, adipocyte-specific STAT3 knockout increases body weight and leads to spontaneous hepatic steatosis [22]. A global loss of STAT3 impairs glucose homeostasis and leads to hepatic steatosis [23]. Interestingly, the PPARG agonist Lobe mitigates hepatic steatosis and preadipocyte differentiation induced by STAT3 inactivation, highlighting the role of STAT3 in regulating lipid metabolism via PPARG [24,25]. Additionally, STAT3 activation in adipocytes promotes adipocyte differentiation by directly binding to its target gene C/EBP-β [26]. However, the effects of STAT3 on lipid metabolism in primary duck hepatocytes have not been explored.

Previous research revealed a lack of Thr resulted in alterations in 12 genes in the JAK-STAT pathway and a notable decrease in STAT3 phosphorylation in the liver of ducks [27,28]. Therefore, it was hypothesized that the ability of Thr to reduce hepatic lipid metabolism is regulated by STAT3 phosphorylation. The study aimed to explore how the silencing and dephosphorylation of the STAT3 gene impact lipid deposition in primary duck hepatocytes regulated by Thr. This was achieved by utilizing siRNA and the Stattic inhibitor to test the hypothesis.

## 2. Results

### 2.1. Influence of Thr on Activity and Fat Deposition in Hepatocytes

We first investigated the effect of Thr on primary duck hepatocytes. The findings showed that as Thr levels in the medium rose, cell survival rose significantly (*p* < 0.01), while LDH activity decreased steadily (*p* < 0.01), leveling off at 25 μM Thr. Below 25 μM Thr, hepatocyte damage led to significant cell death (Figure 1A,B). The ORO staining and OD values indicated a significant decrease (*p* < 0.01) in lipid droplet content with increasing levels of Thr. Cellular TG contents first decreased and then increased (*p* < 0.01) with rising Thr levels (Figure 1A,C). TG content was lowest at 50 μM Thr (Figure 1C). In addition, specific mRNA markers associated with lipid synthesis showed increased (*p* < 0.01) expression under conditions of decreased Thr levels. Among the factors identified were Peroxisome proliferator-activated receptor alpha (*PPARA*), Peroxisome proliferator-activated receptor gamma (*PPARG*), Elongase of very long-chain fatty acids 2 (*ELOVL2*), and Elongase of very long-chain fatty acids 7 (*ELOVL7*) (Figure 1D). Western immunoblot analysis further confirmed that Thr deficiency increased (*p* < 0.01) PPARG protein levels (Figure 1E). These findings collectively indicate a significant enhancement in the lipid accumulation ability of primary duck hepatocytes under Thr deficiency.

### 2.2. Thr Regulates the JAK-STAT Pathway in Primary Duck Hepatocytes

The lack of Thr led to a notable reduction (*p* < 0.01) in the mRNA expression levels of Platelet-derived growth factor receptor beta (*PDGFRB*), Interleukin-6 (*IL-6*), Janus kinase 1 (*JAK1*), and STAT3. In contrast, the mRNA expression levels of Janus kinase 2 (*JAK2*), Tyrosine kinase 2 (*TYK2*), Signal transducer and activator of transcription 1 (*STAT1*), and Signal transducer and activator of transcription 5B (*STAT5B*) were notably elevated (*p* < 0.01) (Figure 2A). Levels of STAT3 protein and its phosphorylation were significantly decreased (*p* < 0.01) when Thr was absent or deficient in the medium (Figure 2B).

### 2.3. Transcriptome Analysis at Different Thr Levels

To explore the relevant pathways and potential target genes of Thr regulation in hepatic lipid deposition, RNA-seq was conducted on primary duck hepatocytes incubated with 0 and 25 μM Thr for 36 h. The principal component analysis (PCA) plot displayed distinct clusters between 0 and 25 μM Thr groups (Figure 3A). A total of 3556 DEGs were identified, with 1540 upregulated and 2016 downregulated in the 0 μM Thr group (Figure 3B,C). KEGG pathway analysis revealed that these DEGs were significantly enriched in the PI3K-Akt signaling pathway, the regulation of TRP by inflammatory mediators, ECM–receptor interaction, the cell cycle, arachidonic acid metabolism, linoleic acid metabolism, steroid biosynthesis, glycerophospholipid metabolism, and the PPAR signaling pathway (Figure 3D). GO pathways functional enrichment analysis highlighted the involvement of Thr in cellular processes and metabolic pathways (Figure 3E). In line with prior findings, notable alterations in various JAK-STAT pathway genes were noted, with a significant decrease in STAT3 (*p* < 0.01), specifically in the Thr-deficient group (Figure 3F). Additionally, some key gene mRNA levels in the PPAR-related pathway also underwent significant changes (*p* < 0.05) in the absence of Thr, such as those of *PPARG*, *PPARA*, and Matrix metalloproteinase 1 (*MMP1*) (Figure 3G). The changes in mRNA expression of Prostaglandin-endoperoxide synthase 2 (*PTGS2*), Phospholipase A2 Group IVE (*PLA2G4E*), Phospholipase A2 Group V (*PLA2G5*), and Phospholipase A2 Group IIA (*PLA2GL2A*) in the arachidonic acid metabolic pathway are also noteworthy (Figure 3H). RT-PCR was finally utilized to identify the presence of Matrix metalloproteinase 1 (*MMP1*) and Matrix metalloproteinase 13 (*MMP13*) at various levels of Thr, confirming the findings from RNA-seq (Figure 3I). These findings underscore the regulatory role of Thr in metabolic signaling pathways, particularly in cell metabolism and lipid synthesis.

### 2.4. Thr Deficiency-Induced Increase in Lipid Deposition Is Caused by Reduction in STAT3 Levels

Transfection with siRNA-STAT3 significantly reduced (*p* < 0.01) the levels of STAT3 mRNA, protein, and phosphorylation in primary duck hepatocytes (Figure 4A,B,F). This indicated the successful establishment of the knockdown model. STAT3 knockdown increased (*p* < 0.01) lipid droplet amounts and TG contents at different Thr levels (Figure 4C,D). In addition, STAT3 knockdown increased (*p* < 0.01) mRNA expressions of *ELOVL7*, *PPARG*, and *PPARA* at varying Thr levels, but did not affect (*p* = 0.970) *ELOVL2* mRNA expression (Figure 4E). STAT3 knockdown also increased (*p* < 0.01) PPARG protein expression across different Thr levels (Figure 4G).

### 2.5. Transcriptome Analysis after STAT3 Knockdown

To further explore the downstream genes regulating lipid metabolism with STAT3, RNA-seq was performed to determine the genes and pathways influenced by STAT3 knockdown in hepatocytes. The PCA plots exhibited good reproducibility within each group (Figure 5A). A total of 16,340 genes were identified, with 1832 DEGs in the siRNA-STAT3 group compared to the NC group, of which 912 genes were downregulated and 920 were upregulated (Figure 5B,C). In the volcano plot, several genes with similar trends to Thr in lipid metabolism regulation were identified, including *PPARG*, *ELOVL7*, *MMP1*, *PTGS2*, *PLA2G4E*, *PLA2G5*, *PLA2GL2A*, and Fatty acid binding protein 3 (*FABP3*) (Figure 5C). KEGG analysis revealed a notable increase in pathways, including arachidonic acid metabolism, ether lipid metabolism, and linoleic acid metabolism. Furthermore, the inhibition of STAT3 affected various pathways including the PI3K-Akt signaling pathway, mTOR signaling pathway, apoptosis, growth hormone synthesis, secretion, the activity and regulation of inflammatory mediators on TRP channels, and interaction with ECM receptors (Figure 5D). By conducting Venn analysis on the two RNA-seq datasets, 650 DEGs were identified as common to both datasets (Figure 5E). An analysis of KEGG enrichment showed that the shared DEGs were predominantly enriched in ECM–receptor interaction, the MAPK signaling pathway, apoptosis, and in protein digestion and absorption (Figure 5F). The gene interaction network indicated that among these DEGs, STAT3 directly regulated genes such as *PTGS2*, *MMP1*, and *MMP13* (Figure 5G).

### 2.6. Thr Deficiency Promotes Triglyceride Accumulation through Inactivation of STAT3

To confirm whether Thr affects lipid deposition by activating STAT3, hepatocytes were treated with Stattic (a specific inhibitor of p-STAT3). As expected, Stattic significantly decreased (*p* < 0.05) the phosphorylation levels of STAT3-Y706 and STAT3-S728 in liver cells without affecting the mRNA (*p* = 0.686) and protein (*p* = 0.767) levels of STAT3 (Figure 6A,D). And, with Stattic incubation, Thr levels no longer influenced (*p* = 0.640) the cellular TG content (Figure 6B,C). Following this, it was discovered that the phosphorylation of STAT3 significantly elevated the mRNA levels of *MMP1*, *MMP13*, Tissue inhibitor of metalloproteinase 4 (*TIMP4*), *ELOVL7*, and *PPARG*, while decreasing the mRNA levels of *PTGS2* (*p* < 0.01) as shown in Figure 6D,E. In addition, the inhibition of STAT3 phosphorylation enhanced (*p* < 0.01) PPARG protein expression (Figure 6F).

## 3. Discussion

Our data indicate that Thr deficiency increased TG content in primary hepatocytes of duck, and reduced the levels of STAT3 mRNA, protein, and phosphorylation levels, suggesting that the promotion of liver fat accumulation may be partially linked to Thr’s regulation of STAT3. Furthermore, by observing lipid deposition in duck hepatic cells with STAT3 knockdown and dephosphorylation, it was found that reduced STAT3 phosphorylation leads to an increase in triglycerides. The findings back the theory that Thr governs lipid metabolism in duck livers by potentially influencing the phosphorylation of STAT3. Moreover, our research revealed that a lack of Thr leads to higher levels of fat in the liver due to the decreased phosphorylation of STAT3, potentially impacting the production of PPARG protein, *PTGS2*, *ELOVL7*, *MMP1*, *MMP13*, and *TIMP4* mRNA. The findings from the aforementioned study enhance the comprehension of Thr regulation in duck lipid metabolism and establish a scientific foundation for preventing the production of hepatic lipid metabolism disorders in waterfowl through nutrition.

Thr plays a crucial role as a nutrient and signaling molecule, controlling lipid metabolism in both laboratory and living settings [29]. Extensive in vivo research has generated substantial information on how Thr reduces lipid synthesis [30,31]. Gene expression changes are commonly considered intrinsic factors leading to nutritional lipid metabolism disorders [32]. In both previous studies and this research, we consistently found that Thr deficiency in the diet increases triglyceride content in the early growth stage of Pekin ducks’ livers, and affects the levels of STAT3 mRNA, protein, and phosphorylation [28]. This suggests that the JAK-STAT pathway is essential in controlling lipid accumulation caused by a lack of Thr. Comprising three components, the JAK-STAT pathway includes a receptor that receives signals related to tyrosine kinase, the JAK tyrosine kinase that transmits these signals, and the transcription factor STAT that generates effects [33]. The JAK-STAT membrane receptors are diverse and can receive stimuli from various cytokines and growth factor ligands [18]. In this study, Thr deficiency not only reduced *IL6* mRNA, increased PDGFB mRNA, it also altered the levels of its receptors *IL6R*, *PDGRFA*, and *PDGRFB* mRNA, indicating that IL6 and PDGFB may be the primary factors receiving Thr signals at the cell membrane. The connection between ligands and receptors induces the activation of JAK [34]. Likewise, in this research, we discovered a positive association between Thr levels and *JAK1* mRNA, as well as a negative correlation with *JAK2* and *TYK2* mRNA levels. When JAK is activated, it causes Stat proteins to form dimers, move to the nucleus, and control the expression of target genes [35]. We observed that Thr deficiency reduced *STAT3* mRNA, protein, and phosphorylation levels, while increasing *STAT1* and *STAT5B* mRNA levels, suggesting that STAT3 may form homodimers for the subsequent transcriptional regulation of genes. However, additional investigation is required to confirm these hypotheses. Overall, a lack of Thr hinders the initiation of the JAK-STAT pathway, diminishing the propagation of the feedback-controlled signals *SOCS1* and *SOCS2* mRNA, potentially causing irregular downstream gene expression associated with lipid metabolism and lipid metabolism disorders.

To understand the downstream regulatory mechanisms of STAT3, we explored genes and signaling pathways affected by Thr deficiency or STAT3 knockdown in primary duck hepatocytes using RNA-seq. The results revealed significant changes in pathways related to unsaturated fatty acid metabolism, such as with arachidonic acid, linolenic acid, and glycerophospholipids, in both Thr deficiency and STAT3 knockdown conditions, and also affected the expression of PI3K-AKT, cAMP, and MAPK signaling pathways, which are involved in energy transduction and lipid metabolism [36,37,38,39]. TG synthesis is regulated by multiple factors, including fatty acid hydrolysis and polyunsaturated fatty acid biosynthesis [40]. Recent studies have shown that the PLA2 family mainly catalyzes the hydrolysis of phosphatidylcholine to release unsaturated fatty acids [41,42]. Enzymes from the ELOVL families catalyze fatty acid desaturation and carbon chain elongation [43,44]. The PTGS family alleviates excessive fat deposition by converting arachidonic acid into prostaglandins [45]. Our research indicates that Thr deficiency and STAT3 knockdown increase mRNA levels of *PLA2G4E*, and *ELOVL7*, while decreasing mRNA levels of *PLA2G5* and *PTGS2*, suggesting that Thr deficiency promotes long-chain fatty acid synthesis and reduces its decomposition ability through STAT3, which may explain the previously observed increase in C20 and C22 in Thr-deficient duck liver [27]. Previous studies have shown that the overexpression of PTGS2 is mediated by STAT3, and the inactivation of STAT3 is accompanied by the decreased expression of PTGS2 in hepatocellular carcinoma [46]. Our research has also reached the same conclusion. There are currently no reports on the regulation of PLA2 and ELOVL by STAT3, but it is worth noting that we have observed changes in PPARG in Thr-deficient and STAT3 knockdown cells. PPARG, as one of the most important transcription factors in lipid metabolism, enhances the processing of free fatty acids (FFA) to TG storage by regulating many genes, including PLA2 and ELOVL [44,47,48,49]. Existing studies have shown that pSTAT3 (Y705) interacts with the PPARG promoter, enhancing PPARG activation [24]. Similarly, using the potent STAT3 activator inhibitor Stattic, we confirmed that STAT3 inactivation significantly enhances both the mRNA and protein expression of PPARG. 

Our study also uncovered that Thr deficiency, alongside STAT3 knockdown or inactivation, elevates the expression of *MMP1* and *MMP13* mRNA levels. This supports the growing consensus on the critical regulatory role of *MMPs* genes in lipid metabolism. MMP2 knockout resulted in a drastic reduction in adipocyte lipid content and influenced mature adipocyte differentiation in mice [50]. Similarly, eliminating MMP1 and MMP13 significantly enhanced lipid and glucose metabolism in obese mice [51,52]. Further, STAT3-mediated activation of MMPs has been linked to perichondrium degradation in white adipose tissue [53]. Our findings show a significant increase in *MMP1* and *MMP13* mRNA levels following Thr deficiency, STAT3 knockdown, and inactivation. TIMP, an endogenous inhibitor of MMPs, works alongside MMPs to modulate Extracellular matrix (ECM) remodeling [54,55,56]. Excessive remodeling of the ECM can lead to liver fibrosis and reduce its lipolytic capacity [57,58]. Our data indicate that Thr deficiency, STAT3 knockdown, and inactivation significantly increase *TIMP4* mRNA levels and alter ECM regulatory pathways. This suggests that Thr deficiency-induced STAT3 inactivation leads to excessive ECM remodeling, which may explain the decreased activity of hepatocytes observed in Thr deficiency. Notably, MMPs have been shown to be regulated by PPARG activation, contributing to the improvement of the microenvironment in perivascular adipose tissue [59]. Therefore, there is a complex interaction among STAT3, PPARG, and MMPs, which merits further investigation.

## 4. Materials and Methods

### 4.1. Isolation, Culture, and Thr Treatment of Hepatocyte

Cherry Valley duck eggs were procured from a commercial hatchery, Jiangsu Yike Food Group, located in China. These eggs were laid by ducks of the same breed and age on the same day. The product guarantee value composition analysis of the feed for laying ducks showed the following values: crude protein, ≥18.0%; crude ash, ≤16.0%; crude fiber, ≤7.0%; calcium, 2.5–4.5%; phosphorus, ≥0.6%; sodium chloride, 0.3–0.8%; methionine, ≥0.36%; and moisture, ≤14.0%. We collect 200 eggs at a time for follow-up experiments (excluding trial repetition). The collected duck eggs were immediately placed in a commercial incubator (capacity 38400, Yunfeng, China) for hatching at a temperature of 38 °C and a humidity of 65%. Primary cultured hepatocytes from meat duck embryos were prepared using a modified version of a previously established method [60]. Briefly, the livers from 18-day-old (E18) duck embryos were harvested and rinsed multiple times with cold phosphate-buffered saline (PBS) lacking magnesium and calcium. The livers were then minced, cleared of impurities, and subjected to lysis using collagenase type IV (SolarBio, Beijing, China) at 0.2 mg/mL. The digested cells were dispersed by pipetting and filtered through 74 and 38 μm strainers to remove large debris. The cell suspension was centrifuged three times at 800 rpm for 5 min using a centrifuge. Erythrocyte lysate (SolarBio, Beijing, China) was used after removing erythrocyte impurities from the cell suspension. Finally, the cell pellet was ultimately resuspended in a complete medium made up of M199 medium (BasalMedia, Shanghai, China) with the addition of 10% FBS (Gibco, Auckland, New Zealand) and 2% antibiotics (Gibco, Grand Island, NY, USA). The hepatocytes were seeded at 1 × 10^6^ cells per well in a 6-well plate and incubated in a humidified chamber at 37 °C with 5% CO_2_, and, after 24 h, the complete medium was replaced with different levels of Thr medium. The different levels of Thr medium are prepared by diluting customized Thr-free M199 basal medium (BasalMedia, code X099A1, Shanghai, China), and M199 medium (Thr content 250 mg/mL). After 36 h of incubation, morphological characteristics of liver cells were assessed using a microscope (Olympus, Tokyo, Japan), and the cells were collected for subsequent assays.

### 4.2. Transfection with Short Interfering RNA (siRNA) and Stattic Treatment

The siRNA used in the present study were designed based on the duck-derived STAT3 gene sequences in GenBank (XM_038168717.1). Lipofectamine 2000 reagent (Thermo, Waltham, MA, USA) was used to transiently transfect cells with STAT3 siRNA (sense: 3′-GGUACAACAUGCUGACCAATT-5′, antisense: 3′-UUGGUCAGCAUGUUGUACCTT’) and control siRNA (sense: 3′-GGUACAACAUGCUGACCAATT-3′, antisense: ACGUGACACGUUCGGAGAATT). The transfection was conducted according to the manufacturer’s instructions, with 1 μL of Lipo2000 per 2000 pmol of siRNA in OptiMEM medium (Thermo, Waltham, MA, USA) to prepare STAT3-specific siRNA or NC (negative control) siRNA. The design sequence of siRNA-STAT3 was compared in BLAST and synthesized by GenPharm Co., Ltd. (Shanghai, China). Following a 24 h incubation period, cells were exposed to varying concentrations of threonine for 36 h before being harvested for analysis.

The Stattic (MCE, Monmouth Junction, NJ, USA) used in this study is a STAT3 phosphorylation-specific inhibitor. When used, 2 μL of 20 mM Stattic was added to the 6-well plate hepatocytes, incubated for 1 h [61], and then cells received different levels of Thr treatment and were collected for index determination. Dimethyl sulfoxide (DMSO) was used as the negative control during the experiment. 

### 4.3. Cell Viability Assays

The viability of hepatocytes treated with different levels of Thr was evaluated using the CCK-8 Cell Kit (Vazyme, code no A311-01, Nanjing, China). Experiments were performed according to the manufacturer’s instructions. Briefly, 100 μL of cells were seeded in a 96-well plate (Corning, Corning, NY, USA) and cultured in the complete medium for 24 h. Then, the medium was replaced with Thr-containing medium at various levels, and the cells were incubated for 36 h. Cell activity was assessed by incubating with 10 μL of CCK-8 solution for 3 h, and the OD (absorbance) was measured at 450 nm using a microplate reader (TECAN, Rosemont, IL, USA).

### 4.4. Lactate Dehydrogenase (LDH) Content

LDH activity was analyzed using a commercial kit (Nanjing Jiancheng Bioengineering Institute, code no A020-2-2, Nanjing, China) as per the kit’s instructions. Specifically, the cell supernatant was collected and added with the reaction solution to a 96-well plate, and incubated at 37 °C for 20 min. The absorbance values were detected at 450 nm. To normalize LDH activity across different Thr treatments, adjustments were made based on cell activity.

### 4.5. Oil Red O (ORO) Staining

Lipid droplets in hepatocytes were stained using the ORO staining kit (Solarbio, code no G1260, Beijing, China) according to the provided instructions. Briefly, cells were fixed with 4% formaldehyde, stained with ORO solution and hematoxylin, and then observed under a microscope for photography. To quantify ORO staining, the dye was extracted from the cells using isopropanol, and absorbance was measured at 500 nm. Adjustments were made for ORO absorbance values based on cell viability data to account for Thr treatment variability.

### 4.6. Determination of Triglyceride (TG) Content

TG content in hepatocytes was quantified using the recommended protocol by Applygen (code no E1025-105, Beijing, China). Briefly, the cell samples were lysed and denatured at 70 °C. The absorbance of the obtained supernatant was analyzed using a microplate spectrophotometer (Bio-Rad, Hercules, CA, USA). The protein concentration in lysed cells was determined with a BCA protein detection kit (Beyotime, code no P0009, Beijing, China). TG content adjustments were made per mg of protein concentration.

### 4.7. RNA Isolation, Reverse Transcription, and Quantitative Real-Time PCR

Total RNA was isolated from primary duck hepatocytes using the TRIzol reagent (Takara, Dalian, China). Genomic DNA was eliminated, and reverse transcription was conducted according to the manufacturer’s protocol (Takara, Dalian, China). The resulting cDNA was then used for PCR amplification with an Applied Biosystems 7500 Real-Time PCR System (Life Technologies, Carlsbad, CA, USA), utilizing the PowerUp™ SYBR™ Green master mix (Thermo, Waltham, MA, USA). Primer sequences for each target gene are detailed in Table 1, with β-actin employed as the reference gene. Relative gene expression levels were quantified using the 2^−ΔΔCt^ method.

### 4.8. Western Blotting and Immunoprecipitation

The protein lysis buffer consisted of RIPA lysis buffer (Beyotime, Beijing, China), 1% phenylmethanesulfonylfluoride or phenylmethylsulfonyl fluoride (PMSF) (Beyotime, Beijing, China), and 1% phosphorylase inhibitor (Solarbio, Beijing, China). A BCA kit was used to determine the protein concentration in the cell lysate. Proteins were denatured using sodium dodecyl sulfate (SDS) buffer (Beyotime, Beijing, China) and separated on 8–16% polyacrylamide gels (GenScript, Nanjing, China) at 120 V for 60 min. Proteins were then transferred (220 mA, 90 min) to a polyvinylidene fluoride membrane (Bio-Rad, Hercules, CA, USA). After blocking for 30 min in rapid blocking solution (NCM, Suzhou, China), the membrane was incubated with the primary antibody overnight at 4 °C. This was followed by 2 h incubation with a diluted secondary antibody at room temperature. Protein bands were visualized using enhanced chemiluminescent substrates (NCM, Suzhou, China) in a Western blot detection system (Tanon, Shanghai, China) and quantified with ImageJ software (version V1.54g). Details on antibodies are provided in Table 2.

### 4.9. Transcriptome Analysis

Primary duck hepatocytes treated with Thr (0–25 μM) and siRNA-STAT3 interference were processed with TRIzol, and the obtained total cellular RNA was used for RNA-seq analysis. The extended protocol details and data analysis procedures were provided by GENE DENOVO (Guangzhou, China). High-quality reads were aligned to the duck reference genome (https://www.ncbi.nlm.nih.gov/datasets/genome/GCF_015476345.1/ accessed on 8 October 2023) using Bowtie2. Genes with fold changes of 1.5 or greater and *p*-values below 0.05 were deemed statistically significant and identified as differentially expressed genes (DEGs). These DEGs were then analyzed for gene ontology (GO) and KEGG pathway enrichment based on their annotated functions.

### 4.10. Statistical Analysis

Data are expressed as the mean ± standard error of the mean (S.E.M.) of examined parameters. All experiments had multiple biological replicates with a sample size of 3–8. Statistical analyses were performed using one-way ANOVA for the data from different Thr treatments, and two-way ANOVA for the data from Thr treatment and STAT3 interference or inactivation, using the General Linear Model (GLM) procedure in SAS 9.4 (SAS Institute Inc., Cary, NC, USA). Differences among means were tested by the Duncan method. Data with a significance level of *p* < 0.05 were deemed statistically significant. Bioinformatic analysis (PCA, Heatmap, Volcano Plot, GO, and KEGG enrichment Scatter Plot) was performed using the OmicStudio tools at https://www.omicstudio.cn/tool accessed on 10–25 August 2023. String (https://cn.string-db.org/ accessed on 22 August 2023) was used to visualize protein regulatory network interactions.

## 5. Conclusions

Threonine is an essential amino acid for ducks. Our in vitro experiments demonstrated that threonine deficiency inhibited the activity of primary duck hepatocytes and reduced triglyceride accumulation. More importantly, different expression patterns of STAT3 play a critical role in threonine-regulated lipid deposition. The reduction and inactivation of STAT3 lead to lipid metabolic disorders and promote triglyceride accumulation by upregulating the mRNA levels of *ELOVL7*, *PPARG*, *MMP1*, *MMP13*, and *TIMP4*, and downregulating the mRNA level of *PTGS2*. This study comprehensively elucidates the molecular mechanism by which threonine regulates lipid synthesis and metabolism in the duck liver through STAT3, providing a basis for the nutritional prevention of hepatic lipid metabolic disorders.

## Figures and Tables

**Figure 1 ijms-25-08142-f001:**
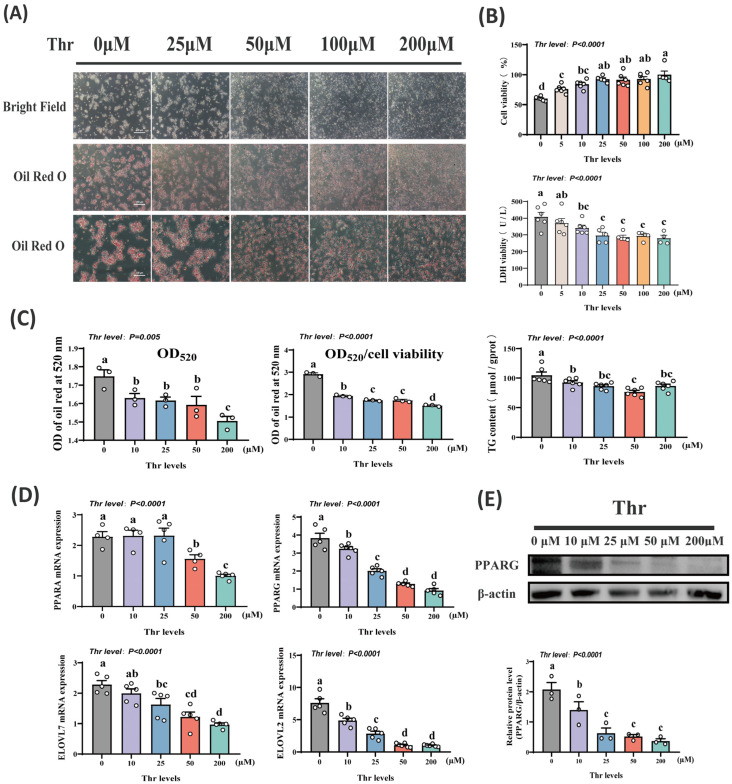
Thr levels influence duck hepatocyte activity and lipid content. (**A**) Hepatocyte morphology and ORO staining with different Thr levels. Scale bars, from top to bottom 200 μm, 200 μm and 100 μm. (**B**) Cell viability and LDH activity detection. (**C**) ORO staining OD values and TG content detection. (**D**) Detection of *PPARA*, *PPARG*, *ELOVL2*, and *ELOVL7* mRNA levels by qPCR. (**E**) Western blotting to detect PPARG protein level. Data are means ± SEM; n = 3–6 biologically independent replicates obtained from independent experiments (represented by white circles). a–d Different letters indicate significant differences between groups (*p* < 0.05).

**Figure 2 ijms-25-08142-f002:**
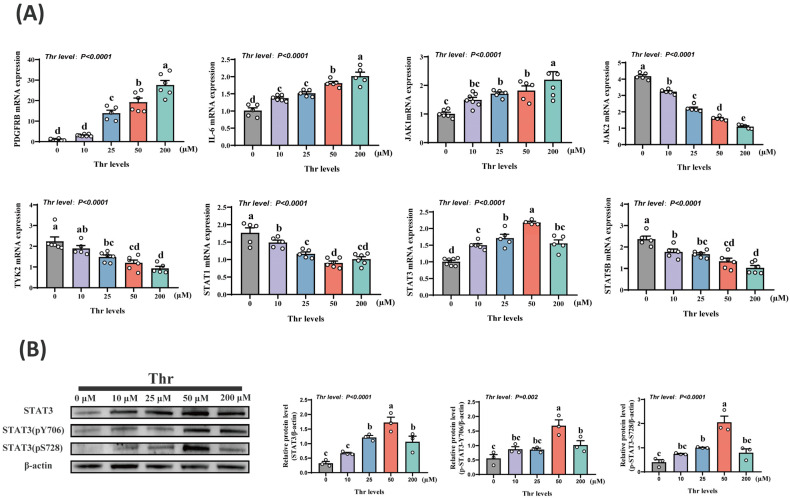
Thr affects the JAK-STAT pathway. (**A**) qPCR detection of *PDGFRB*, *IL-6*, *JAK1*, *JAK2*, *TYK2*, *STAT1*, *STAT3*, and *STAT5B* mRNA levels. (**B**) Western blotting to detect STAT3 protein and phosphorylation level. Data are means ± SEM; n = 3–6 biologically independent replicates obtained from independent experiments (represented by white circles). a–e Different letters indicate significant differences between groups (*p* < 0.05).

**Figure 3 ijms-25-08142-f003:**
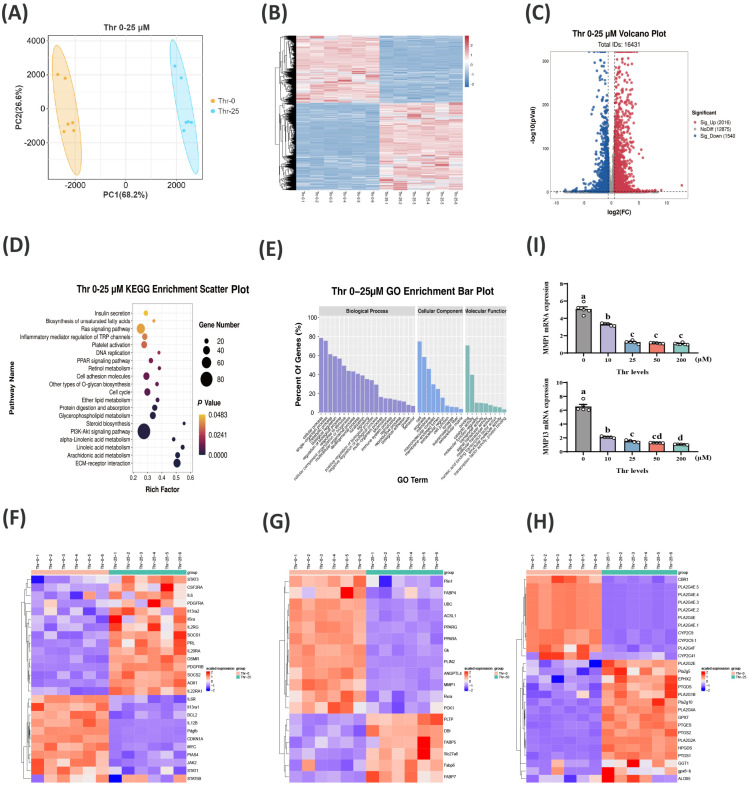
Transcriptomic data highlight Thr regulation of lipid metabolism. (**A**) PCA of RNA-seq data after treatment with 0 and 25 μM Thr. (**B**) Differential gene heat map. (**C**) Volcano plot of differential genes. (**D**) Top twenty KEGG pathways enrichment plot of differential genes. (**E**) GO enrichment analysis. (**F**) JAK-STAT signaling pathway differential gene heatmap. (**G**) PPAR signaling pathway differential gene heatmap. (**H**) Arachidonic acid metabolism pathway differential gene heatmap. (**I**) qPCR validation of sequencing data. Data are means ± SEM; n = 3–6 biologically independent replicates obtained from independent experiments (represented by white circles). a–d Different letters indicate significant differences between groups (*p* < 0.05).

**Figure 4 ijms-25-08142-f004:**
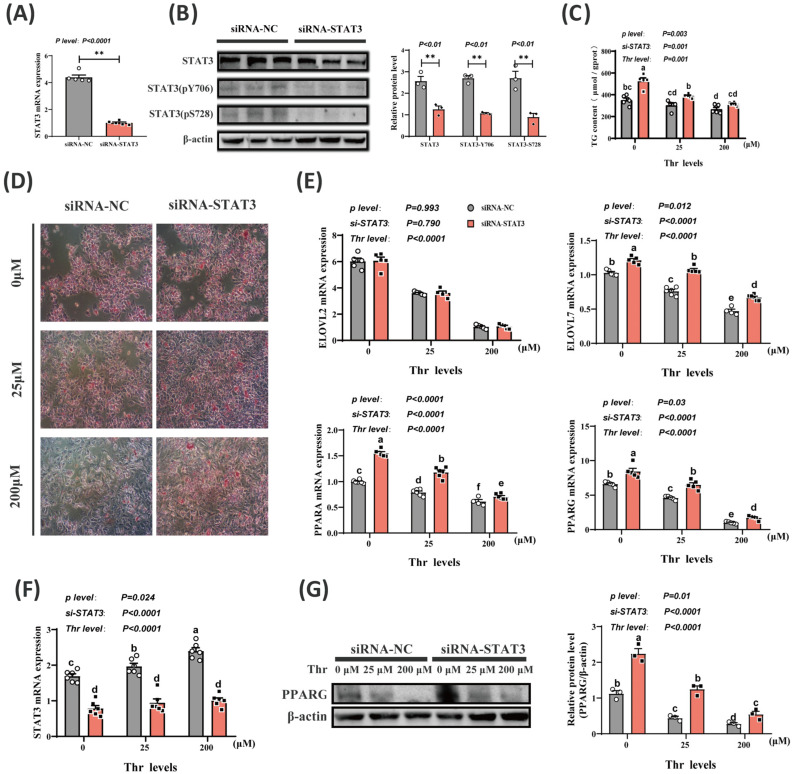
Thr regulates lipid deposition in hepatocytes through STAT3. (**A**,**B**) STAT3 knockdown efficiency assay. qPCR and Western blotting detection of STAT3 gene expression and protein levels after siRNA transfection. (**C**,**D**) Triglyceride content and ORO staining of cells under different Thr levels after STAT3 knockdown. Scale bars, 100 μm. (**E**,**F**) qPCR detection of *STAT3*, *ELOVL2*, *ELOVL7*, *PPARA*, and *PPARG* mRNA expression levels in cells under different Thr levels after STAT3 knockdown. (**G**) Western blotting detection of PPARG protein in cells under different Thr levels after STAT3 knockdown. Data are means ± SEM; n = 3–6 biologically independent replicates obtained from independent experiments (siRNA-NC represented by white circles, siRNA-STAT3 represented by black squares). **, *p* < 0.01. a–f Different letters indicate significant differences between groups (*p* < 0.05).

**Figure 5 ijms-25-08142-f005:**
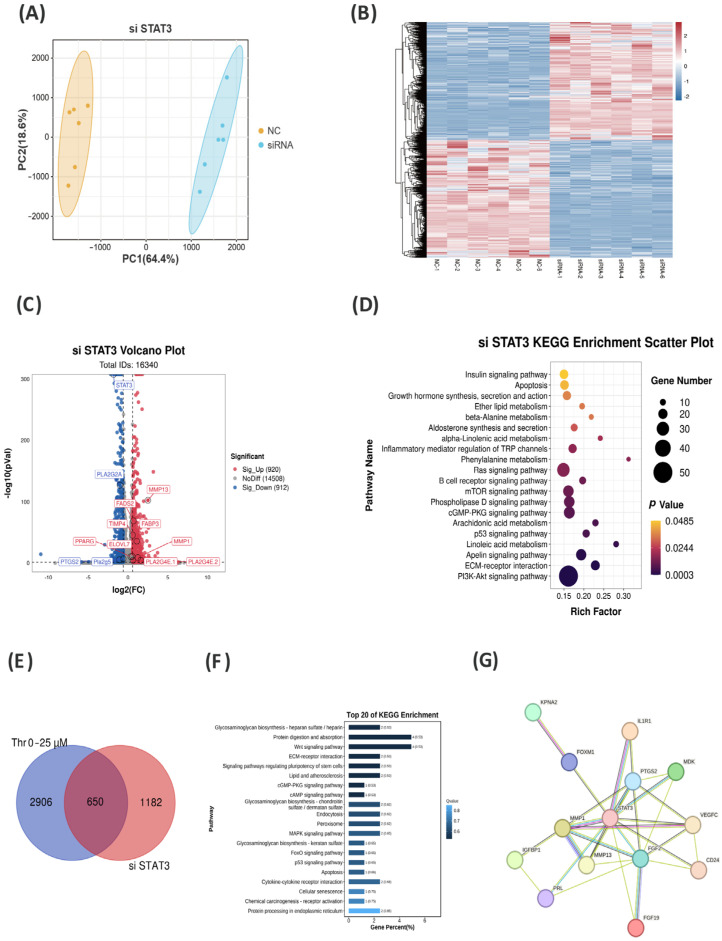
Transcriptomic data reveal key signaling pathways influenced by Thr in regulating lipid metabolism through STAT3. (**A**) PCA of RNA-seq data after treatment with siRNA-NC and siRNA-STAT3. (**B**) Heatmap of differentially expressed genes. (**C**) Volcano plot of differentially expressed genes. (**D**) Enrichment plot of the top twenty KEGG pathways for differentially expressed genes. (**E**) Overlapping genes between different Thr levels of RNA-seq and siSTAT3 RNA-seq data. (**F**) KEGG enrichment of overlapping genes. (**G**) Regulatory network between overlapping genes (different colored circles) and STAT3, the green line represents gene neighborhood, red line represents gene fusions, blue represents gene co-occurrence, yellow line represents text mining, black line represents co-expression, lavender line represents protein homology, cyan line represents curated databases, and magenta represents experimentally determined relationships.

**Figure 6 ijms-25-08142-f006:**
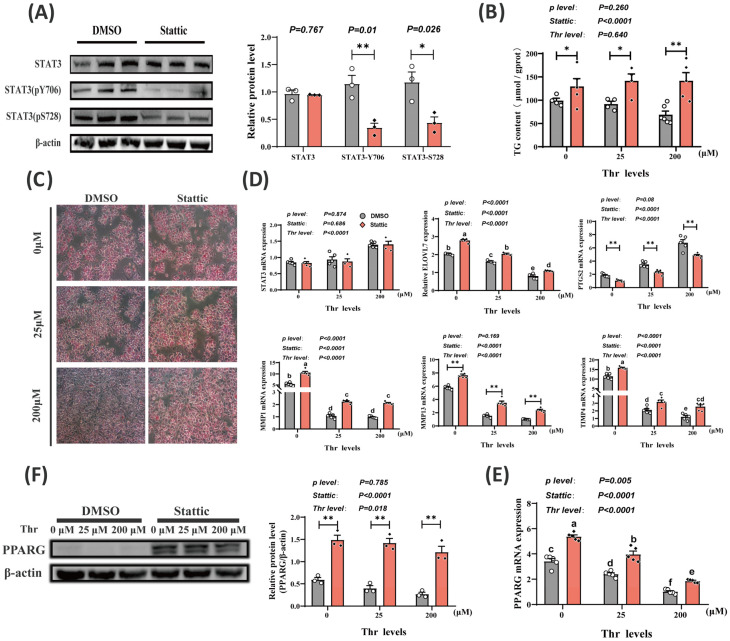
Thr regulates lipid deposition in hepatocytes through STAT3 activation. (**A**) Western blotting detects STAT3 dephosphorylation efficiency. (**B**,**C**) Triglyceride content and ORO staining of cells under different Thr levels after STAT3 inactivation. Scale bars, 100 μm. (**D**,**E**) qPCR detection of *STAT3*, *ELOVL7*, *MMP1*, *MMP13*, *TIMP4*, *PTGS2*, and *PPARG* mRNA expression in cells at different Thr levels after STAT3 inactivation. (**F**) Western blotting detection of expression and phosphorylation of PPARG protein in cells at different Thr levels after STAT3 dephosphorylation. Data are means ± SEM; n = 3–6 biologically independent replicates obtained from independent experiments (siRNA-NC represented by white circles, siRNA-STAT3 represented by black squares). *, *p* < 0.05; **, *p* < 0.01. a–f Different letters indicate significant differences between groups (*p* < 0.05).

**Table 1 ijms-25-08142-t001:** Primer sequences for PCR amplification.

Genes ^1^	Sequence (5′-3′)	Product Length (5′-3′)
PPARA	F: ACCAGCATCCAGTCCTTCATCCA	146 bp
	R: AACCTTCACAAGCATGTACTCCGTAA	
PPARG	F: CCCAAGTTTGAGTTCGCTGT	192 bp
	R: GCTGTGACGACTCTGGATGA	
ELOVL2	F: CCAAGGTGCTGTGGTGGTAT	162 bp
	R: CGCAGGGTATCCAGTTCAGG	
ELOVL7	F: AGCACTGGTTACCTTGCCTC	174 bp
	R: GCGTGTGTGCCCTTAACAA	
PDGFRB	F: CCACCCATGCCTCCGATGAAATC	131 bp
	R: CCTCTTCCTGTAGCAATCCACCAAG	
IL-6	F: CGTGTGCGAGAACAGCATG	364 bp
	R: GTCTCGGAGGATGAGGTG	
JAK1	F: CTGTGCAGATACGATCCAGAAGGTG	101 bp
	R: TTCTTGAGGTCAGCGATGTGATTCC	
JAK2	F: ACCTATTTGCACAGTGGCGAGATG	139 bp
	R: AGTGGTGTTTGGTCCCTTTCTTTGG	
TYK2	F: TCAACATCGGCAAGGACACCAAC	80 bp
	R: CTCGTTCATCCCGTGCCAGTTC	
STAT1	F: ACGCAGGAAGCAGAACGAATGAG	139 bp
	R: TTTGAGATCACGACAATGGGAAGGG	
STAT3	F: AGGAGGAGGCGTTTGGGAAGTAC	135 bp
	R: CGATGGTGTTGCTGAAGGAGGTG	
STAT5B	F: TCACCGACATCATCTCTGCCCT	212 bp
	R: ATTGCGGGTGCTTTCGTTCTT	
MMP1	F: GCCTACACGGACCCCAATGA	195 bp
	R: CAGCATGTATCTGCCCTTGAAGA	
MMP13	F: ATGGAAGCAGGCTACCCCAG	162 bp
	R: AGTGTGCAGGACACGGACAA	
TIMP4	F: TCTGCGATTCTGCTTTAGTGAT	354 bp
	R: TACAGCCACAGCCCATTTGA	
PTGS2	F: GAAAAACCACGACCAGGTGC	198 bp
	R: ACAGCCTTTCACGTTGTTGC	
β-actin	F: ATGTCGCCCTGGATTTCG	165 bp
	R: CACAGGACTCCATACCCAAGAA	

^1^ PPARA: Peroxisome proliferator-activated receptor alpha. PPARG: Peroxisome proliferator-activated receptor gamma. ELOVL2: Elongase of very long-chain fatty acids 2. ELOVL7: Elongase of very long-chain fatty acids 7. PDGFRB: Platelet-derived growth factor receptor beta. IL-6: Interleukin-6. JAK1: Janus kinase 1. JAK2: Janus kinase 2. TYK2: Tyrosine kinase 2. STAT1: Signal transducer and activator of transcription 1. STAT3: Signal transducer and activator of transcription 3. STAT5B: Signal transducer and activator of transcription 5b. MMP1: Matrix metalloproteinase 1. MMP13: Matrix metalloproteinase 13. TIMP4: Tissue inhibitor of metalloproteinase 4. PTGS2: Prostaglandin-endoperoxide synthase 2. β-actin: Beta-actin.

**Table 2 ijms-25-08142-t002:** Details of antibodies.

Antibodies ^1^	Cat No.	Source	Dilution of WB
STAT3	A1192	ABclonal, Wuhan, China	1:1000
p-STAT3(Y706)	bs-22386R	Bioss, Beijing, China	1:1000
p-STAT3(S728)	AF3294	Affinity, Cincinnati, OH, USA	1:1000
PPARG	ab178860	abcam, Cambridge, UK	1:1000
β-actin	NB600-532	Novus, Colorado, USA	1:5000
HRP-labeled Goat Anti-rabbit IgG(H+L)	ab205718	abcam, Cambridge, UK	1:10,000

^1^ PPARG: Peroxisome proliferator-activated receptor gamma. STAT3: Signal transducer and activator of transcription 3. β-actin: Beta-actin.

## Data Availability

The RNA-seq data from this study had been deposited in the National Center for Biotechnology Information with the accession numbers PRJNA1089378. All datasets generated or analyzed during this study are available from the corresponding author on reasonable request.

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
