# Peer review of "Threonine Deficiency Increases Triglyceride Deposition in Primary Duck Hepatocytes by Reducing STAT3 Phosphorylation"

_ijms, 2024, doi:10.3390/ijms25158142_

Round 1

Reviewer 1 Report

Comments and Suggestions for Authors

The purpose of this study was to investigate the effects of threonine supplementation on the reduction of triglyceride deposition in duck liver cells by up-regulating the phosphorylation of STAT3, thereby improving the disorder of liver lipid metabolism.However, the readability of the manuscript can be greatly improved to better communicate the importance of your research. After editing and some revisions, I feel that the manuscript is fit for publication. Important weaknesses that need to be satisfactorily addressed are listed below.

1.     Line 42 Please replace the balanced amino acid with the balanced amino acid level.

2.     Line 44 Keep the names consistent in context and replace all poultry with waterfowl.

3.     Line 44  What are limiting amino acids? What is the result of a lack of limiting amino acids? Please add clarification.

4.     Line 49  Why choose Pekin ducks liver? Is it possible to choose another breed of duck liver? Are there differences and limitations in metrics? Please elaborate on the reasons.

5.     Line 59  Can only be treated with cytokines? Can it be treated with the growth factors mentioned earlier? Please keep it consistent.

6.     Line 60  Replace the high-fat diet in the article with the low-protein diet mentioned in the previous article. Be consistent before and after.

7.     Line 64    What is preadipocyte differentiation? Please explain in detail.

8.     Line 144  Please replace all duck liver cells in the article with primary duck liver cells. Please keep proper nouns consistent throughout the text.

9.     Line 292  Are MMP and MMPS the same? If the same, please be consistent in the full text, if you do not sympathize with additional explanations.

Author Response

Dear Editor and Reviewers,

We are very grateful to you for reviewing our manuscript and providing detailed comments. Those comments and suggestions are all valuable and very helpful for revising and improving our paper, as well as the important guiding significance to our researches. Moreover, we have studies comments carefully and have finished the corrections. We hope it meets with approval. The main corrections in the paper and the responds to your comments are as follows.

(Original comments are in italic, we reply to the question highlighted with red color, the added context and significant change in revised manuscript highlighted with yellow in revised manuscript.)

Response to Reviewer 1 Comments

Point 1: Line 42 Please replace the balanced amino acid with the balanced amino acid level.

Response 1: Thank you very much for your suggestions. We have already changed "the balanced amino acid" to "the balanced amino acid levels" as detailed in line 42 of the manuscript.

Point 2: Line 44 Keep the names consistent in context and replace all poultry with waterfowl.

Response 2: Thank you very much for your suggestions. We have already changed "poultry" to "waterfowl" in the manuscript.

Point 3: Line 44 What are limiting amino acids? What is the result of a lack of limiting amino acids? Please add clarification.

Response 3: Thanks for your kindly comment. We understand that limiting amino acids refer to those that cannot be synthesized within the animal's body and must be obtained from the diet. The consequences of lacking these amino acids include impaired growth, compromised protein synthesis, disrupted immune function, and metabolic pathway disturbances, particularly in lipid metabolism[1]. We have already added and modified this section in line 44 of the manuscript.

  1. Yap YW, Rusu PM, Chan AY, Fam BC, Jungmann A, Solon-Biet SM, Barlow CK, Creek DJ, Huang C, Schittenhelm RB, Morgan B, Schmoll D, Kiens B, Piper MDW, Heikenwälder M, Simpson SJ, Bröer S, Andrikopoulos S, Müller OJ, Rose AJ. Restriction of essential amino acids dictates the systemic metabolic response to dietary protein dilution. Nat Commun. 2020 Jun 9;11(1):2894. doi: 10.1038/s41467-020-16568-z. PMID: 32518324; PMCID: PMC7283339.

Point 4: Line 49 Why choose Pekin ducks liver? Is it possible to choose another breed of duck liver? Are there differences and limitations in metrics? Please elaborate on the reasons.

Response 4: Thanks for your kindly comment. Pekin ducks are one of the largest commercially raised duck breeds in China[2]. This popularity ensures easier access to a large number of animals for research purposes compared to less common duck breeds. Additionally, Pekin ducks are known for their robust growth and efficient feed conversion, making them suitable for studies related to growth performance, nutrient utilization, and metabolism. Their liver physiology and lipid metabolism may be well-studied, providing a comparative baseline in research. Therefore, we believe that the liver of the Pekin duck is a better research subject in China for studying lipid metabolism in waterfowl livers. As for the livers of other duck breeds, we believe there are no significant differences or limitations in indicators. This is because we have found that Threonine has similar effects on lipid metabolism in both mammals and birds[3], and there is higher homology within duck species. Therefore, we consider Threonine to have an equal impact on the livers of all duck breeds.

  1. Chen X, Shafer D, Sifri M, Lilburn M, Karcher D, Cherry P, Wakenell P, Fraley S, Turk M, Fraley GS. Centennial Review: History and husbandry recommendations for raising Pekin ducks in research or commercial production. Poult Sci. 2021 Aug;100(8):101241. doi: 10.1016/j.psj.2021.101241. Epub 2021 May 14. PMID: 34229220; PMCID: PMC8261006.
  2. Chen J, Qian D, Wang Z, Sun Y, Sun B, Zhou X, Hu L, Shan A, Ma Q. Threonine supplementation prevents the development of fat deposition in mice fed a high-fat diet. Food Funct. 2022 Jul 18;13(14):7772-7780. doi: 10.1039/d2fo01201d. PMID: 35766226.

Point 5: Line 59 Can only be treated with cytokines? Can it be treated with the growth factors mentioned earlier? Please keep it consistent.

Response 5: Thank you very much for your suggestions. We have reexamined the relevant literature and found that the main regulators of STAT3 are cytokines. Therefore, we have changed "growth factors or cytokine" to "cytokines" in line 59 of the manuscript.

Point 5: Line 60 Replace the high-fat diet in the article with the low-protein diet mentioned in the previous article. Be consistent before and after.

Response 6: Thank you very much for your suggestions. We consider a high-fat diet and a low-protein diet to be two different types of diets. A high-fat diet typically involves consuming a large amount of fat without restricting carbohydrate and protein intake. A low-protein diet refers to a diet with reduced protein content. This type of diet primarily limits protein sources. These two types of feed are different, and therefore, it may not be necessary to make changes to the manuscript

Point 7: What is preadipocyte differentiation? Please explain in detail.

Response 7: Thank you very much for your suggestions. We have reviewed related articles and provided an explanation of preadipocytes differentiation. Adipogenesis is a complex process whereby the multipotent adipose-derived stem cell is converted to a preadipocyte before terminal differentiation into the mature adipocyte. Preadipocytes are present throughout adult life, exhibit adipose fat depot specificity, and differentiate and proliferate from distinct progenitor cells. The authors clearly define preadipocytes as an adipose-lineage committed cell population destined to proliferate and differentiate only into mature adipocytes. A fundamental property of preadipocytes is their ability to constantly proliferate and differentiate into mature adipocytes, allowing adipose tissue to maintain its functional plasticity and expansion throughout the lifespan of the host [4].

  1. Sarantopoulos CN, Banyard DA, Ziegler ME, Sun B, Shaterian A, Widgerow AD. Elucidating the Preadipocyte and Its Role in Adipocyte Formation: a Comprehensive Review. Stem Cell Rev Rep. 2018 Feb;14(1):27-42. doi: 10.1007/s12015-017-9774-9. PMID: 29027120.

Point 8: Line 144 Please replace all duck liver cells in the article with primary duck liver cells. Please keep proper nouns consistent throughout the text.

Response 8: Thank you very much for your suggestions. We have already changed " duck liver cells" to "primary duck liver cells" throughout the article, and highlighted the changes in yellow in the revised manuscript

Point 9: Line 292 Are MMP and MMPS the same? If the same, please be consistent in the full text, if you do not sympathize with additional explanations.

Response 9: Thank you very much for your suggestions. 'MMP' and 'MMPs' are the same in the article, so we have changed 'MMP' to 'MMPs' throughout the article, and highlighted the changes in yellow in the revised manuscript.

In addition, we rechecked the content of the article and found some obvious errors, such as in the results section “2.3 Transcriptome Analysis at Different Thr Levels”, where there are some issues with Figure 3. The “4.10 Statistical Analysis” section in Materials and Methods is also incorrect, and we have made corrections to these

We tried our best to improve the manuscript and made some changes in the manuscript. We appreciate for Editors/Reviewers’ warm work earnestly and hope that the correction will meet with approval. Once again, thank you very much for your comments and suggestions.

Best regards,

Yong Jiang

Reviewer 2 Report

Comments and Suggestions for Authors

Dear authors this study provides new and important data about the role of threonine on triglyceride deposition in duck hepatocytes. The use of the English language was also appropriate. However, in the materials and methods section authors must describe more analytically the number of eggs, and the diets that were followed because the reasons behind the described effects will be understood better. Moreover, authors must answer why they used only one reference in the relative genes expression analysis and they have to describe under the table1., the genes more analytically by writing their names. Finally, the conclusion, is too short and did not describe the impact of this work in the appropriate way. For these reasons, the present study must be major revised to be published in the International Journal of Molecular Sciences. All the comments are described also section by section below.

Abstract

-----------------------

Results

-----------------------

Discussion

-----------------------

Material and Methods

 4.1. Isolation, Culture, and Thr Treatment of Hepatocyte

Authors must describe more analytically the diets used, and the number of eggs which were taken.

4.7. RNA Isolation, Reverse Transcription, and Quantitative Real-Time PCR

Please, write the full names of the evaluated genes in table 1. Authors must also answer why they used only one reference gene for the relative genes’ expression evaluation.

Conclusion

The conclusion must be more analytical and accurate, it is too short. Please rewrite it.

Author Response

Dear Editor and Reviewers,

We are very grateful to you for reviewing our manuscript and providing detailed comments. Those comments and suggestions are all valuable and very helpful for revising and improving our paper, as well as the important guiding significance to our researches. Moreover, we have studies comments carefully and have finished the corrections. We hope it meets with approval. The main corrections in the paper and the responds to your comments are as follows.

(Original comments are in italic, we reply to the question highlighted with red color, the added context and significant change in revised manuscript highlighted with yellow in revised manuscript.)

Response to Reviewer 2 Comments

Point 1: Material and Methods

4.1. Isolation, Culture, and Thr Treatment of Hepatocyte

Authors must describe more analytically the diets used, and the number of eggs which were taken.

Response 1: We are thankful to the respected reviewer for pointing out this problem. Due to our unclear description, you may not have fully understood our isolation process. We extracted duck livers from unhatched duck embryos rather than hatched ducklings. Therefore, the duck embryos did not have related feed, but we provided information on the daily feed for egg-laying ducks in the manuscript. Additionally, we reiterated the number of duck eggs taken for our experiments. In our study, 2×10^8 primary duck liver cells were isolated from 200 duck eggs at 18-day-old duck embryos, which is equivalent to using 200 six-well plates. This provided the necessary quantity of cells for one experiment (excluding experimental replicates.

Point 2: 4.7. RNA Isolation, Reverse Transcription, and Quantitative Real-Time PCR

Please, write the full names of the evaluated genes in table 1. Authors must also answer why they used only one reference gene for the relative genes’ expression evaluation.

Response 2: Thank you very much for your suggestions. We have added the full names of all genes in Table 1. In addition, the reasons for using a single reference gene to assess the expression of related genes are as follows:

  1. Selecting a widely recognized stable reference gene, such as β-actin, enhances the consistency and reproducibility of experimental results.The β-actin gene we selected is widely recognized as a housekeeping gene and has been extensively used in lipid synthesis and metabolism studies, where it is employed as a single reference gene[1-3]. Therefore, using β-actin as a reference gene is crucial for comparing results between different laboratories and ensuring result replication.
  2. Using one reference gene can save costs, considering that quantitative detection is not inexpensive, and we aim to obtain accurate results with minimal expenditure. We found that using one reference gene to evaluate the expression of related genes yielded results consistent with transcriptome data, confirming the accuracy of our results.
  3. Introducing multiple reference genes may increase the risk of experimental variation, particularly in sample handling and PCR reactions. A stable reference gene can help reduce the potential impact of these variations on experimental outcomes.

  1. Li Q, Zhao Y, Guo H, Li Q, Yan C, Li Y, He S, Wang N, Wang Q. Impaired lipophagy induced-microglial lipid droplets accumulation contributes to the buildup of TREM1 in diabetes-associated cognitive impairment. Autophagy. 2023 Oct;19(10):2639-2656. doi: 10.1080/15548627.2023.2213984. Epub 2023 May 19.
  2. Wang X, Wu R, Liu Y, Zhao Y, Bi Z, Yao Y, Liu Q, Shi H, Wang F, Wang Y. m6A mRNA methylation controls autophagy and adipogenesis by targeting Atg5 and Atg7. Autophagy. 2020 Jul;16(7):1221-1235. doi: 10.1080/15548627.2019.1659617. Epub 2019 Aug 26.
  3. Wang L, Chen Y, Wei J, Guo F, Li L, Han Z, Wang Z, Zhu H, Zhang X, Li Z, Dai X. Administration of nicotinamide mononucleotide improves oocyte quality of obese mice. Cell Prolif. 2022 Nov;55(11):e13303. doi: 10.1111/cpr.13303. Epub 2022 Jul 10.

Point 3: The conclusion must be more analytical and accurate, it is too short. Please rewrite it.

Response 3: We are thankful to the respected reviewer for pointing out this problem. Based on the suggestions, we have revised the conclusion section of the article accordingly. The specific modifications are as follows.

Threonine is an essential amino acid for ducks. Our in vitro experiments demonstrated that threonine deficiency inhibited the activity of primary duck hepatocytes and reduced triglyceride accumulation. More importantly, different expression patterns of STAT3 play a critical role in threonine-regulated lipid deposition. The reduction and inactivation of STAT3 lead to lipid metabolic disorders and promote triglyceride accumulation by upregulating the mRNA levels of ELOVL7, PPARG, MMP1, MMP13, and TIMP4, and downregulating the mRNA level of PTGS2. This study comprehensively elucidates the molecular mechanism by which threonine regulates lipid synthesis and metabolism in the duck liver through STAT3, providing a basis for the nutritional prevention of hepatic lipid metabolic disorders.

We tried our best to improve the manuscript and made some changes in the manuscript. We appreciate for Editors/Reviewers’ warm work earnestly and hope that the correction will meet with approval. Once again, thank you very much for your comments and suggestions.

Best regards,

Yong Jiang

Round 2

Reviewer 1 Report

Comments and Suggestions for Authors

Good!

Reviewer 2 Report

Comments and Suggestions for Authors

Dear authors, all the appropriate changes that were spotted have been made. The present work is ready to be published in the IJMS journal.